# High-Spatial-Resolution NDVI Reconstruction with GA-ANN

**DOI:** 10.3390/s23042040

**Published:** 2023-02-11

**Authors:** Yanhong Zhao, Peng Hou, Jinbao Jiang, Jiajun Zhao, Yan Chen, Jun Zhai

**Affiliations:** 1School of Earth Science and Mapping Engineering, China University of Mining and Technology, Beijing 100083, China; 2Satellite Environment Application Center, Ministry of Ecology and Environment, Beijing 100094, China; 3Chinese Research Academy of Environmental Sciences, Beijing 100012, China

**Keywords:** reconstruction algorithm, NDVI, high spatial resolution, GA-ANN, MODIS, Landsat

## Abstract

The normalized differential vegetation index (NDVI) for Landsat is not continuous on the time scale due to the long revisit period and the influence of clouds and cloud shadows, such that the Landsat NDVI needs to be filled in and reconstructed. This study proposed a method based on the genetic algorithm–artificial neural network (GA-ANN) algorithm to reconstruct the Landsat NDVI when it has been affected by clouds, cloud shadows, and uncovered areas by relying on the MODIS characteristics for a wide coverage area. According to the self-validating results of the model test, the RMSE, MAE, and R were 0.0508, 0.0557, and 0.8971, respectively. Compared with the existing research, the reconstruction model based on the GA-ANN algorithm achieved a higher precision than the enhanced spatial and temporal adaptive reflectance fusion model (ESTARFM) and the flexible space–time data fusion algorithm (FSDAF) for complex land use types. The reconstructed method based on the GA-ANN algorithm had a higher root mean square error (RMSE) and mean absolute error (MAE). Then, the Sentinel NDVI data were used to verify the accuracy of the results. The validation results showed that the reconstruction method was superior to other methods in the sample plots with complex land use types. Especially on the time scale, the obtained NDVI results had a strong correlation with the Sentinel NDVI data. The correlation coefficient (R) of the GA-ANN algorithm reconstruction’s NDVI and the Sentinel NDVI data was more than 0.97 for the land use types of cropland, forest, and grassland. Therefore, the reconstruction model based on the GA-ANN algorithm could effectively fill in the clouds, cloud shadows, and uncovered areas, and produce NDVI long-series data with a high spatial resolution.

## 1. Introduction

The NDVI is one of the important indicators that are commonly used to evaluate and monitor dynamic vegetation changes [1] and land cover changes [2], as well as to conduct vegetation phenology analyses [3,4] and ecological environment monitoring [5]. The long observation period of remote-sensing satellites and the defects of remote-sensing technology, coupled with the influence of clouds, cloud shadows, and other factors [6], can result in a discontinuity of the NDVI on the time scale. However, in order to achieve accurate monitoring, predictions, and estimations [7,8], the requirements of researchers for the spatial resolution of data are constantly improving [9]. To solve this problem, the idea is to adopt the method of data fusion to realize the complementary advantages of data.

By summarizing the existing literature, there were mainly three types of methods found: weight-based methods, unmixing-based methods, and machine learning-based methods [10,11]. Among the weight-based methods, the most representative methods are the spatial and temporal adaptive reflectance fusion model (STARFM) [12,13], the enhanced spatial and temporal adaptive reflectance fusion model (ESTARFM) [14], and the spatial and temporal reflectance fusion method based on the unmixing theory and a fuzzy C-clustering model (FCMSTRFM) [15]. The latter two methods were improved based on the STARFM. To a certain extent, these models effectively improved the performance of reflectance in disturbed and non-uniform regions [15], but still did not completely solve the influence of clouds on the reflectance, nor did they monitor mutations (that is, NDVI values that are very different from the surrounding area or the same location during the same period in adjacent years, which may be caused by natural disasters such as floods, heavy precipitation, and so on). Unmixing-based methods such as the space–time data fusion method [16], the space and time reflectance unmixing model [17], and the flexible space–time data fusion algorithm (FSDAF) [18] have also been widely used for their high computational efficiency. In terms of mutation event monitoring, the accuracy of predictions was improved [19], but the models ignored the mixed pixels in the construction process [20]. Therefore, their pixel accuracy still needed to be further improved. The sparse representation-based spatiotemporal reflectance fusion model [21], enhanced single-pair learning-based reflectance fusion algorithm [22], error-bound-regularized sparse coding dictionary-learning model [23], multilayer extreme learning machine [24], and other machine-learning algorithms have been widely used due to their high precision.

The weight-based and unmixing-based data fusion methods have a large uncertainty in the case of complex land use, and cannot predict mutations in the NDVI promptly. The machine-learning data fusion model can achieve high-precision results with its strong learning ability. However, the GA-ANN algorithm was not used for Landsat and MODIS vegetation index conversion or for filling in missing data values in existing studies. In this study, a new method was developed for reconstructing a high-spatial-resolution NDVI by directly establishing the relationship between the MODIS and Landsat NDVI pixels. It uses the GA-ANN data fusion method to give full play to the advantages of the MODIS vegetation index product on the spatial scale to reconstruct the clouds, cloud shadows, and uncovered areas in the Landsat NDVI.

## 2. Study Area and Dataset Preprocessing

### 2.1. Study Area

This paper selected Beijing–Tianjin–Hebei as the study area, which is an important core area of northern China’s economy. It consists of two cities, Beijing and Tianjin, and the Hebei province. Its land cover is dominated by cropland, forest, grassland, artificial surfaces, and town areas, as shown as Figure 1. There are many types of complex land cover in the region, and it presents significant heterogeneity. The climate is the temperate monsoon type, and it is characterized by high temperatures and rain in the summer due to the effect of ocean water vapor, and a cold and dry climate in winter due to the inland cold air.

### 2.2. Dataset and Data Preprocessing

The dataset used in this study included MODIS data, Landsat data, and Sentinel data, and the characteristics of the dataset are shown in Table 1.

MODIS vegetation index product data (MOD13Q1, http://ladsweb.nascom.nasa.gov/data/search.html, accessed on: 11 April 2022) with a spatial resolution of 250 m and a temporal resolution of 16 d were used in this study. MODIS’s quality control band (DetailedQA) performed quality control on the vegetation index and provided support for the pixel removal of clouds and cloud shadows. The Landsat8 L2 dataset provided the corrected reflectance data; it contained five visible and near-infrared bands and two short-wave infrared bands. In this study, red and near-infrared bands were selected to calculate the NDVI, and the quality control band (QA_PIXEL) was used to control the quality of the data so as to remove clouds and cloud shadows and reduce the influence of clouds and cloud shadows on the pixels.

The Sentinel-2 L2 dataset was an orthoimage bottom-of-atmosphere (BOA) corrected reflectance product, and it included 12 bands: an aerosols band, three visible light bands (blue, green, and red), an NIR band, four vegetation red edge bands, a water vapor band, and two short-wave infrared bands. The resulting NDVI data obtained after calculating the reflectance of the red and near-infrared bands of Sentinel-2 were selected to verify the 30 m NDVI product we obtained. The time and space resolutions were 5 d and 10 m, respectively.

The data preprocessing mainly included cloud and cloud shadow removal, time and space matching, and projection coordinate system conversion. In addition to the above basic preprocessing steps, the MOD13Q1 and Sentinel-2 data also needed to be resampled to 30 m.

Due to the difference in the temporal and spatial resolution between MODIS and Landsat, it was necessary to match time and space and transform the projection coordinate system. First of all, we matched the time. According to the data introduction of MOD13Q1, the corresponding relationship between the MOD13Q1 and the actual time can be determined. The actual time refers to the time from the beginning to the end of the MOD13Q1 data observation. The starting time of the MOD13Q1 was January 1st of each year, that is, the data of period 001 was synthesized from the maximum value from 00:00:00 on 1 January to 23:59:59 on 16 January, and marked as 1 January in the MOD13Q1. Therefore, the Landsat data and the Sentinel data needed to be unified with the real-time of the MODIS. Secondly, the spatial resolution of both the MOD13Q1 and the produced NDVI using the Sentinel satellite data needed to be resampled to 30 m. At the same time, the unification of the projected coordinate system was also necessary, and we unified it to the WGS84 coordinate system.

The reflectivity of the cloud pollution area (area affected by clouds and cloud shadows) was abnormal, and the NDVI was lower than that in the unpolluted area [25,26]. To effectively remove the abnormal effect on reflectivity, it was essential to remove the cloud (include cirrus) [27] pollution pixels from the remote-sensing satellite images. Therefore, we needed to remove the clouds (including cirrus) and cloud shadows. The above operations were implemented on the GEE platform with the help of quality control bands.

## 3. Methodology

### 3.1. NDVI Calculation

Vegetation indices based on visible and near-infrared bands have been widely used in vegetation detection and health assessments [28], among which the NDVI is most commonly used in vegetation-related monitoring. The definition of the NDVI is shown by Formula (1) [29]:(1)$$\mathrm{NDVI}=\frac{\rho_{NIR}-\rho_{RED}}{\rho_{NIR}+\rho_{RED}}$$
where $\rho_{RED}$ is the reflectance in the red band and $\rho_{NIR}$ is the reflectance in the near-infrared band.

Different remote-sensing images have different NDVI formulae. The NDVI formulae for Landsat 8 L2 and Sentinel-2 L2 in this study are shown in Table 2. The NDVI for MODIS comes from the NDVI band of the MOD13Q1. The range of the NDVI is [−1,1]. NDVI < 0 indicates that the ground cover is cloud, water, snow, etc.; NDVI=0 indicates that there are rocks or bare soil; and NDVI < 0 indicates that there is vegetation coverage, and the value increases with an increase in the coverage. The valid values of the NDVI were filtered, and only the pixel values with NDVI ≥ 0 were kept.

### 3.2. Reconstructing NDVI Model Based on GA-ANN Algorithm

The Landsat NDVI’s spatial resolution is 30 m, but its revisit time is too long, and it is affected by clouds and cloud shadows, making the NDVI abnormal. Existing studies have shown that it is feasible to use the MODIS NDVI to reconstruct missing Landsat NDVI values [30]. Therefore, this study took advantage of the MODIS coverage to fill the Landsat’s NDVI for the cloud-polluted (pixels affected by clouds or cloud shadows) and non-covered areas. To begin with, pixel pairs between the MODIS and Landsat NDVIs that were not polluted by clouds were screened (one pair of pixels are pixels with same time and position in the Landsat and MODIS after data processing) to establish the corresponding relationship between the MODIS and Landsat NDVI pixels. Then, with the MODIS NDVI and the corresponding pixels as input data, the Landsat NDVI areas with cloud-polluted or missing pixels were reconstructed. Finally, the accuracy of the reconstructed NDVI was evaluated and the NDVI was retrained until the accuracy requirements were met. The method flowchart is shown in Figure 2.

In existing studies, linear models [30] have been commonly used to describe the differences between MODIS and Landsat. However, a linear model is too simple and cannot adequately describe the relationship between MODIS and Landsat NDVIs. Therefore, this study used the GA-ANN algorithm to establish the correspondence between the Landsat NDVI and the NDVI of MOD13Q1. 

Artificial neural network (ANN) algorithms have been widely used in data fusion [31]. A genetic algorithm (GA) is a method based on natural gene selection [32], which is more suitable for describing the actual situation of the correspondence between pixels in data fusion. The genetic algorithm–artificial neural network (GA-ANN) is formed by introducing a genetic algorithm (GA) based on an artificial neural network (ANN) model [33]. The GA-ANN algorithm has been widely used in inversions of various parameters [34,35] due to its excellent inversion accuracy. Simultaneously, the GA-ANN model is better suited to the complex characteristics of land use types. Therefore, the GA-ANN algorithm was applied to reconstruct the Landsat NDVI. The main parameters of the GA-ANN are shown in Table 3.

In this study, we randomly and uniformly selected 1000 random points in each pair of Landsat–MODIS pixels (23 pairs per year, time range of 2018–2020) to realize the MODIS–Landsat NDVI pixel correspondence. As far as the GA-ANN algorithm, the structure of the GA-ANN model includes an input layer, a hidden layer, and an output layer. The input data were the MODIS NDVI data, and the output data were the Landsat NDVI data, so the input layer and output layer were set to 1. As shown in Figure 3, a different hidden layer of the reconstructed NDVI model based on the GA-ANN had differences in the RMSE and R. When the hidden layer was set to 5, the model had the highest R and lowest RMSE. 

The realization of reconstructing the model mainly depended on the Google Earth Engine (GEE) platform and MATLAB r2018b software (provided by the American MathWorks Corporation in Natick, MA, USA). The data preprocessing and random point selection were implemented on the GEE platform, and the nonlinear pixel correspondence between the MODIS and Landsat, the establishment of the model, and the production of the dataset were dependent on MATLAB. The MOD13Q1, Landsat reflectivity, and Sentinel-2 reflectivity data all needed to be preprocessed.

### 3.3. The Other Existing Reconstruction Models

The ESTARFM is a typical weight-based method that improves the accuracy of the predicted fine-resolution reflectance and performs well in heterogeneous regions [36]. It is based on the premise that the changes in the reflectance of the fine resolution and coarse resolution images are consistent [15]. The ESTARFM includes six main steps: (1) the intersection of two Landsat NDVI images was obtained by using the threshold segmentation method, and the adjacent pixels similar to the central pixel of the window were selected as similar pixels; (2) the weight of the similar pixels were calculated through the spectrum and distance difference between the predicted pixel and similar pixels; (3) the conversion coefficient of similar pixels was calculated through a regression analysis between the Landsat and MODIS data; (4) the weight values and conversion coefficients were combined based on two pairs of Landsat and MODIS pixel values, and a preliminary fused image for the prediction time was generated; and (5) finally, the final-moment fusion result was generated through the weight function [36]. This method needed two pairs of MODIS and Landsat images to be prepared with a matching time and position, and one MODIS image at a predicted time.

The FSDAF is based on a spectral unmixing analysis and thin plate spline (TPS) interpolation, which is not only suitable for heterogeneous landscapes, but also for predicting gradient changes [37]. The FSDAF includes six main steps: (1) classify the fine-resolution image at t_1_; (2) estimate the temporal change of each class in the coarse-resolution image from t_1_ to t_2_; (3) predict the fine-resolution image at t_2_ using the class-level temporal change and calculate the residuals at each coarse pixel; (4) predict the fine-resolution image from the coarse image at t_2_ with a TPS interpolator; (5) distribute residuals based on the TPS prediction; and (6) obtain the final prediction of the fine-resolution image using information in the neighborhood [37]. This method needs one pair of MODIS and Landsat images to be prepared with a matching time and position, a classification map of the Landsat image, and one MODIS image at the predicted time.

The ESTARFM and FSDAF are widely used for data reconstruction because of their high accuracy in complex land use types. To verify the accuracy of the reconstruction algorithm, the ESTARFM and FSDAF were selected and compared with the methods presented in this study.

### 3.4. Accuracy Assessment

An accuracy assessment is mainly quantitatively evaluated from the difference and correlation degree between the predicted image data and the real image data. 

In terms of the pixel differences, the accuracy assessment of the results of existing methods mainly included the mean absolute error (MAE), average absolute error (AAE), mean square error (MSE), root mean square error (RMSE), and other indicators to quantitatively evaluate the difference between the predicted value and the true value [38,39,40]. Among them, the MAE is less affected by outliers than the MSE, and can better represent the distribution of normal data. The MAE can also better reflect the actual error between the predicted value and the real value than the AAE. In addition, the RMSE and MAE are typical performance measures that indicate the difference between expected and actual values.

In terms of images and pixel correlation, the coefficient of determination (R^2^), the correlation coefficient (R), the peak signal-to-noise ratio (PSNR), the structural similarity index measure (SSIM), and other indexes are usually used to describe whether two sets of data are correlated or similar [41,42,43]. Among them, the SSIM is used to measure the similarity of two images. It ranges from 0 to 1, and the larger the image, the more similar it is. The PSNR is a metric for evaluating image quality. Its unit is dB, and the larger its value, the less image distortion there is. R^2^ is generally used to assess the degree of agreement between the predicted value and the actual value. R is used to describe the degree of linear correlation between two variables; if R>0, there is a positive correlation; if R=0, there is no correlation; and if R<0, there is a negative correlation.

In this study, the RMSE, MAE, R, PSNR, and SSIM were selected as the accuracy assessment indicators to evaluate the differences and the correlation degree of the predicted value and the true value. They can be expressed as follows: (2)$$\mathrm{RMSE}=\sqrt{\frac{\sum_{i=1}^{n}{\left(\mathrm{NDV}I_{{\mathrm{pre}}_{i}}-\mathrm{NDV}I_{tru_{i}}\right)}^{2}}{n}}$$
(3)$$\mathrm{MAE}=\frac{\sum_{i=1}^{n}\left|\mathrm{NDV}I_{pre_{i}}-\mathrm{NDV}I_{tru_{i}}\right|}{n}$$
(4)$$R=\sqrt{1-\frac{\sum_{i=1}^{k}{\left(X_{i}-Y_{i}\right)}^{2}}{\sum_{i=1}^{k}{\left(Y_{i}-\bar{Y}\right)}^{2}}}$$
(5)$$\mathrm{PSNR}=10\cdot log_{10}\left(\frac{max_{pre}^{2}}{\mathrm{MSE}}\right)$$
(6)$$\mathrm{SSIM}=\frac{\left(2\mu_{pre}\mu_{tru}\right)\left(2\sigma_{pre\;tru}+c_{2}\right)}{\left(\mu_{pre}^{2}+\mu_{tru}^{2}+c_{1}\right)\left(\sigma_{pre}^{2}+\sigma_{tru}^{2}+c_{2}\right)}$$
where n is the number of pixels; $\mathrm{NDV}I_{pre_{i}}$ is the NDVI predicted by the model; $\mathrm{NDV}I_{tru_{i}}$ is the true NDVI; k is the number of scenes (32 scenes in a year); $X_{i}$ is the mean of the predicted NDVI for scene *i*; $Y_{i}$ is the mean of the true NDVI for scene *i*; $\bar{Y}$ is the annual mean of the true NDVI; $max_{pre}$ is the maximum of the predicted NDVI; MSE is the mean square error between the original image and the predicted image, where $\mathrm{MSE}=\frac{1}{n}\overset{n}{\underset{i=1}{\sum }}{\left(\mathrm{NDV}I_{tru}-\mathrm{NDV}I_{pre}\right)}^{2}$; $\;\mu_{pre}$ is the mean of the predicted NDVI; $\mu_{tru}$ is the mean of the true NDVI; $\sigma_{pre}^{2}$ is the variance of the predicted NDVI; $\sigma_{tru}^{2}$ is the variance of the true NDVI; $\sigma_{pre\;tru}$ is the covariance of the predicted NDVI and the true NDVI; and $c_{1}={\left(k_{1}L\right)}^{2}$ and $c_{2}={\left(k_{2}L\right)}^{2}$ are constants, where L is the dynamic range of pixels, $k_{1}=0.01$, and $k_{2}=0.03$.

## 4. Results

### 4.1. Performance of Reconstructed NDVI Model

#### 4.1.1. The Self-Validation of Reconstructed NDVI Model

The self-validation of the reconstructed NDVI model can effectively evaluate the feasibility of the model. To achieve model self-validation of the reconstructed NDVI model, we divided the dataset into two parts: the training set (80% of the dataset) and the validation set (20% of the dataset). The training set was used to train and build the model, and the validation set was used to verify the accuracy of the model. The results are shown in Figure 4; the RMSE was 0.0508, the MAE was 0.0557, and R was greater than 0.89. The self-validation results show that the reconstructed NDVI model based on the GA-ANN algorithm could obtain NDVI results with a high spatial resolution and a high accuracy.

#### 4.1.2. Evaluation of Reconstructed NDVI Model

Since the true NDVI was difficult to obtain, this study took the Landsat NDVI without cloud pollution as the real NDVI to verify the authenticity of the reconstructed NDVI model. The difference between the NDVI that was made by reconstructing an NDVI model of sample plots 1, 2, 3, and 4 (predicted NDVI) and the NDVI of Landsat (true NDVI) was compared to verify the authenticity. Simultaneously, the RGB images of the corresponding plots were also used for verification. From the perspective of the accuracy assessment (Table 4), the RMSE of the reconstructed NDVI and the actual Landsat NDVI in the four plots ranged from 0.01 to 0.03, and the RMSE of sample plot 4 was the smallest. The MAE of sample plot 3 was the smallest (0.0198); sample plot 2 had the largest MAE (0.0518). The PSNR and SSIM of the four sample plots were all greater than 15 dB and 0.7, respectively. 

In terms of the visual NDVI differences (Figure 5), the NDVI simulated by the reconstruction based on the GA-ANN was visually consistent with the original Landsat NDVI. It was also visually consistent with true color images in the four sample plots, which proved that the GA-ANN-based reconstruction algorithm could effectively fill in the missing pixels of the NDVI in Landsat. 

#### 4.1.3. Verification with Sentinel Data

To verify whether the reconstruction algorithm was feasible and the accuracy of the result was reasonable, the NDVI predicted by the reconstruction algorithm was verified by the Sentinel NDVI. Therefore, the Sentinel NDVI was taken as the true NDVI and was compared with the reconstructed NDVI using the methods described in this study to analyze the quantitative evaluation of the results. 

The RMSE and MAE between the reconstructed NDVI based on the GA-ANN reconstruction algorithm and the Sentinel NDVI pixels directly indicated the deviation between the two pixels. This study selected three types of vegetation, forest, grassland, and cropland, with obvious NDVI changes for further research, and randomly selected 100 points within their corresponding land use types (see Figure 6a) to verify the accuracy of the reconstructed NDVI. By comparing random points from the Sentinel NDVI with the NDVI reconstructed by the new method in this study, the RMSE of the cropland was found to be about 0.2048 and the MAE was 0.2246; the RMSE of the forest was about 0.2848 and the MAE was 0.2976; and the RMSE of the grassland was about 0.2281 and the MAE was 0.2453. 

The mean NDVI represents the condition of the vegetation development. From the perspective of the mean of random points, the variation trend of the NDVI obtained by the method in this paper was basically the same as that of the Sentinel NDVI, as shown in Figure 6b. However, the range of the predicted NDVI in this paper was significantly different from that of Sentinel; the maximum value of the Sentinel NDVI was about 0.7, while the maximum value of the predicted NDVI was about 0.3. In the cropland part, the maximum NDVI difference between the method in this paper and Sentinel was 0.3596 for [8.28–9.13); in the forest part, the maximum NDVI difference between the method in this paper and Sentinel was 0.4770 for [9.13–9.29); and in the grassland part, the maximum NDVI difference between our method and Sentinel was 0.3970 for [8.28–9.13).

From the perspective of the correlation trend of the NDVI on the time scale, the R values of the means for random points in the land use types of cropland, forest, and grassland on the time scale were all about 0.97 (shown in Figure 6c), indicating that the NDVI predicted by the methods in this paper was similar to the Sentinel NDVI, and that there was a strong correlation.

From the visual differences in images, we focused on analyzing the most distinctive northern forest (1) and southern cropland (2) in the Beijing–Tianjin–Hebei region. During the period from 21 March to 6 April, 2020, we compared the spatial distribution of the cloud-affected true color images, the NDVI filled by the method in this paper, and the Sentinel NDVI, as shown in Figure 7. Clouds and cloud shadows (Figure 7(a-1,a-2)) in the true color images caused pixels to be missed in the Landsat NDVI (Figure 7(b-1,b-2)), and the purpose was to reconstruct these missing pixels through the MODIS NDVI data in this study. The NDVI reconstructed by the GA-ANN was basically consistent with the Sentinel NDVI in spatial distribution. However, the reconstructed NDVI was not as detailed as Sentinel. This was because the difference in the spatial resolution between the two directly led to differences in the spatial detail in the results of the NDVI, as shown in Table 1. The spatial resolution of Sentinel-2 L2 is 10 m, which is much higher than that of the Landsat 8 L2 reflectivity data.

### 4.2. Comparison with Existing Methods

In the Beijing–Tianjin–Hebei region, the land use types are complex with obvious spatial heterogeneity. This remains a key issue to be addressed by data fusion methods. As shown in Figure 1, the vegetation types are mainly cropland, forest, and grassland, and their proportions are 46.48%, 21.1%, and 15.78%, respectively, accounting for 83.36%. The proportion of artificial surfaces and towns is also high, which is 12.67%. Therefore, combined with the main vegetation types in the study area, plots with significant spatial heterogeneity changes (sample plots 1, 2, 3, and 4) were selected as experimental comparison areas. Among them, sample plot 1 was the northern forest, which was dominated by forest and supplemented by grassland; sample plot 2 was near the edge of the city; and sample plot 3 was the western part of the study area, consisting mainly of grassland. Sample plot 4 was mainly cropland in the southern part of Beijing–Tianjin–Hebei. In addition to cropland, it also included many small towns and artificial surfaces. 

Table 5 shows the RMSE and MAE between the NDVIs using the ESTARFM, the FSDAF, and the GA-ANN algorithm with the true Landsat NDVI. Figure 8 shows a visual comparison of the NDVIs simulated by the ESTARFM, the FSDAF, and the reconstruction based on the GA-ANN.

In terms of the error between the predicted NDVI and the true NDVI, the reconstruction based on the GA-ANN had the smallest RMSE compared with that of the ESTARFM and FSDAF for the four sample plots, especially for sample plot 4, where the RMSE was 0.0066. Except for the fact that the MAE of the ESTARFM was the smallest in the forest land use type (sample plot 1), the MAE of the reconstruction algorithm based on the GA-ANN was the smallest in the other sample plots, and the minimum was 0.0198 in sample plot 3. This shows that the reconstruction algorithm based on the GA-ANN had a higher accuracy than the ESTARFM and the FSDAF in different sample plots. In addition to the reconstruction based on the GA-ANN, in the forest and cropland areas (sample plots 1 and 4) where the NDVI changes significantly, the RMSE and MAE of the ESTARFM were better than those of the FSDAF. In grass-dominated sample plot 3, the ESTARFM and FSDAF also had a higher accuracy, the RMSE reached 0.01, and the MAE was less than 0.02. In the city fringe (sample plot 2), the prediction visual effect and pixel accuracy of the ESTARFM and FSDAF were both poor, and the RMSE and MAE were both about 0.2. In comparison, the NDVI predicted by the GA-ANN in the four sample plots were all the closest to the true Landsat NDVI. This proved that the reconstruction based on the GA-ANN was more suitable for NDVI prediction with diverse land use types, and the forecast results were relatively stable. 

From a visual point of view (Figure 8), when comparing the ESTARFM, FSDAF, and reconstructed NDVI based on the GA-ANN, the predicted NDVI images of sample plots 1, 3, and 4 were the most similar to the actual NDVI, while sample plot 2 was quite different. The results show that the ESTARFM and FSDAF overestimated the NDVI of urban fringes. In summary, the reconstructed algorithm performed well both in pixel accuracy and from a visual perspective, and it was applicable to different sample plots of different land use types. The filling effect of the GA-ANN based on the reconstruction of the Landsat NDVI was better than that of the ESTARFM and FSDAF. It had universality and stable prediction accuracy for different sample plots.

## 5. Analysis and Discussion

The Landsat NDVI was discontinuous on the time scale due to clouds, cloud shadows, and the absence of Landsat images. In this regard, it was very necessary to accurately reconstruct the NDVI. Firstly, the pixel pairs of the MODIS products and the Landsat NDVI were used to establish the correlation, and then the MODIS product NDVI data of the missing areas were used as input to reconstruct the missing pixels affected by clouds, cloud shadows, and other affected areas. This study used the Beijing–Tianjin–Hebei region as the study area and proposed a new algorithm to reconstruct the NDVI based on the GA-ANN algorithm. 

### 5.1. Spatiotemporal Analysis of the High-Spatial-Resolution NDVI Product

The NDVI results for the Beijing–Tianjin–Hebei region in 2020 are shown in Figure 9 and Figure 10. On the time scale, the mean value of the NDVI in the Beijing–Tianjin–Hebei region increased slowly with time, and then decreased slightly in the [5.24–6.25) time period. After that, it increased again until [7.27–8.12), when the NDVI began to decrease slowly and continuously (as shown in Figure 9). The trend in the NDVI for the entire study area was consistent with that of the cropland NDVI, which was confirmed by the proportion of land use types in the Beijing–Tianjin–Hebei region (Figure 1). Specifically, the proportion of cropland was 46.48%, accounting for about 50%.

On a spatial scale (Figure 10), the most significant changes in the NDVI were for the northern forest and southern cropland in the Beijing–Tianjin–Hebei area. Among them, starting at [1.1–12.31], the forest in the northern part of the Beijing–Tianjin–Hebei area continuously increased and then decreased until [9.29–10.15), at which point the greenness of the forest had weakened, which was in line with the phenological changes of the forest.

The southern cropland showed two stages of “increase—decreasing”. The first stage was from 1 January. to 25 June; the second stage was from 25 June to 16 November; and until the end of the year, the NDVI basically changed little. Based on the analysis of the crop pattern of “winter wheat and summer maize” in the Beijing–Tianjin–Hebei region [44], “winter wheat” is generally sown in early and middle October and harvested in June of the following year; “summer maize” is generally sown in mid-to-late June and harvested at the end of September. The first stage was consistent with the phenological changes of winter wheat, with the NDVI increasing continuously since January, indicating that the wheat continued to grow after overwintering, and the NDVI was in a state of turning green. Then, the NDVI weakened until after June, indicating that the wheat was in the mature stage in June. The second stage was consistent with the phenological changes of summer maize. After the winter wheat was harvested, the summer corn was sown immediately. With an increase in time, the corn continued to grow and the NDVI increased until the end of September, at which point the NDVI weakened. Then, winter wheat was sown again, and the NDVI did not change much as winter began.

The reconstructed NDVI in the Beijing–Tianjin–Hebei region was consistent with the vegetation changes over time and space, indicating that the reconstructed NDVI was feasible. The results could be applied to describe vegetation condition assessments and analyses in detail. 

### 5.2. Uncertainty of Reconstructed NDVI Model Based on GA-ANN Algorithm

The errors in the reconstructed NDVI model based on the GA-ANN algorithm in this study came from random point selection, data resampling, land use data, and different sensors and bands used by MODIS, Landsat, and Sentinel. The inevitable errors led to limitations of the method.

#### 5.2.1. Influence of Selection of Random Points

The selection of random points was related to the accuracy of the GA-ANN model. This study randomly selected 500, 1000, and 2000 random points for discussion. 

As the number of random points increased, the accuracy indexes showed significant differences (Table 6 and Figure 11). In terms of the MAE, with an increase in random points, the MAE increased. This shows that with an increase in random points, the error in the training points accumulated continuously. It also proves that the selection of random points was not the more, the better. The RMSE decreased with an increase in random points. The RMSE decreased by 0.0015 when the number of random points increased by 1000 from 500, while the RMSE decreased by only about 0.0006 when the number of random points increased by 2000 from 1000. The results show that an increase in random points is beneficial to the improvement of training accuracy, but when the random points exceed 1000, the improvement in the training accuracy is not significant. In terms of R, if the random points were less than 1000, R increased with an increase in random points; if the random points were greater than 1000, R decreased with an increase in random points. When 1000 random points were selected, R reached the maximum value (0.8971). 

An increase in the hidden layer had no obvious effect on the MAE. The increase in the hidden layer made the RMSE fluctuate slightly. The increase in the hidden layer caused fluctuations in the R changes.

#### 5.2.2. Influence of Data Resampling

The resampling of remote-sensing images also brought errors to the results. There are two methods to realize the NDVI pixel correspondence between Landsat and the NDVI: (1) resample the Landsat NDVI to 250 m, and achieve pixel correspondence with the MODIS NDVI (250 m); (2) resample the MODIS NDVI to 30 m, and achieve pixel correspondence with the Landsat NDVI (30 m). This study discussed the reconstructed NDVI model test results of the RMSE and R for the two methods. 

As shown in Figure 12, the accuracy of resampling Landsat to 250 m for model training was better than that of resampling MODIS to 30 m, with a higher R and a smaller RMSE. The R of the Landsat NDVI resampled to 250 m was higher than that of the MODIS NDVI resampled to 30 m by about 0.03; the RMSE of the Landsat NDVI resampled to 250 m was smaller than that of the MODIS NDVI resampled to 30 m by about 0.006. This was because when the MODIS NDVI was resampled to 30 m, it introduced resampling errors and caused errors in the subsequent model training [45].

#### 5.2.3. Influence of Land Use Data and Different Sensors and Bands Used by MODIS, Landsat, and Sentinel

The selection of land use data and the time–space matching of MODIS, Landsat, and Sentinel also caused errors in the verification results.

In this study, considering that the vegetation types in the Beijing–Tianjin–Hebei region are mainly temperate deciduous broad-leaved forests, we directly used the land use type (forest) to represent all forest types. The land use type data we used were LUCC data, which were obtained after manual interpretation based on Landsat remote-sensing images, and the spatial resolution was 30 m. Unlike existing studies [46,47], the land use type data (MCD12Q1) were obtained mainly by MODIS, and the spatial resolution was 250 m. The difference in the selection of land use data may have also led to errors in the validation results. 

In addition to that, errors may have also been derived from the different sensors and band settings carried by MODIS, Landsat, and Sentinel [48]. Space–time matching errors of MODIS, Landsat, and Sentinel may have existed in the pixel correspondence process between MODIS and Landsat and in the verification with Sentinel. The accuracy assessment of the results was dependent on Sentinel-2 L2. The time resolution of Sentinel (5 d) was significantly higher than the 16 d of MODIS products and Landsat, and although we used the maximum synthesis method to generate the Sentinel NDVI for the relevant time period (16 days), the times were not completely consistent and a complete match was not achieved. This was unavoidable, because the time resolutions of the Sentinel and Landsat sensors are not uniform, and the observation time is not exactly the same. 

### 5.3. Advantages of the Reconstructed NDVI Model Based on GA-ANN

Compared with the existing, commonly used data fusion methods (ESTARFM and FSDAF), the reconstructed NDVI model based on the GA-ANN performed better not only for pixel differences, but also in terms of the visual angle. Due to the differences in the method realization mechanisms of the ESTARFM and FSDAF, the ESTARFM required two pairs of images with full coverage of MODIS and Landsat as training units [36]; the FSDAF required one pair of images with full coverage of MODIS and Landsat as the training unit [37]. Therefore, in specific sample plots (sample plots 1 and 4), the NDVI predicted by the ESTARFM was more accurate than that of the FSDAF. It is also worth noting that for the sample plots with complex land use types, such as man-made buildings, the NDVIs predicted by the ESTARFM and FSDAF were significantly different from the true NDVI. In addition, based on the experiments, the data fusion method in this paper preserved the unaffected NDVI data in Landsat to the greatest extent, and used the reconstruction based on the GA-ANN to reconstruct the missing pixels to generate an NDVI dataset. Therefore, in the four sample plots selected in Section 4.2, the NDVI pixel results reconstructed by the reconstruction algorithm had little difference with the actual pixel value. This shows that the reconstruction model was more suitable for sample plots with more complex land use types than the ESTARFM and FSDAF, and the predicted NDVI was more stable and more accurate, especially around cities.

The verification results showed that the difference between the RMSE and MAE of the NDVI reconstructed by the reconstruction algorithm and the Sentinel NDVI were stable for characteristic vegetation types such as forests, grasslands, and cropland. There was a strong correlation between the mean of the NDVI reconstructed by the reconstruction algorithm and the mean of the Sentinel NDVI for the time scale variation. Therefore, the time-series NDVI dataset in the Beijing–Tianjin–Hebei region contributed to high-quality long-sequence vegetation status data and related research.

At the same time, the reconstructed NDVI in the Beijing–Tianjin–Hebei region was consistent with the vegetation changes over time and space, indicating that the reconstructed NDVI is feasible. The results can be applied to describe vegetation condition assessments and analyses in detail.

## 6. Conclusions

The Landsat NDVI was discontinuous on the time scale due to clouds, cloud shadows, and the absence of the Landsat imagery itself. In this regard, it was very necessary to accurately reconstruct the NDVI. Therefore, this study proposed a new reconstruction model to reconstruct the NDVI based on the GA-ANN algorithm and used the MODIS product NDVI as the input to reconstruct the missing pixels affected by clouds, cloud shadows, and other affected areas in the Landsat NDVI data. The MAE, RMSE, and R were used to quantitatively evaluate and test the accuracy metrics of the reconstruction algorithm, where the RMSE was 0.0508, the MAE was 0.0557, and R was above 0.89. Compared with the ESTARFM and FSDAF, the reconstruction model performed well in both pixel accuracy and visual perspectives, and the results also showed that the algorithm was more stable for NDVI predictions and suitable for predictions in different sample plots. In the verification part, the differences in the variation in the random pixel points (RMSE and MAE) between the reconstructed NDVI and the Sentinel NDVI were stable. The R values of the mean random point values for the land use types of cropland, forest, and grassland on the time scale were all about 0.97, and the variation trend and visual differences for the NDVI reconstructed in this study were basically the same as that of the Sentinel NDVI. Therefore, the reconstructed model based on the GA-ANN could effectively and accurately reconstruct the missing pixels in Landsat to generate an NDVI dataset with a spatial resolution of 30 m, which would provide support for high-quality evaluations of vegetation status. The temporal and spatial changes in the reconstructed NDVI products were also consistent with their related phenological changes. 

## Figures and Tables

**Figure 1 sensors-23-02040-f001:**
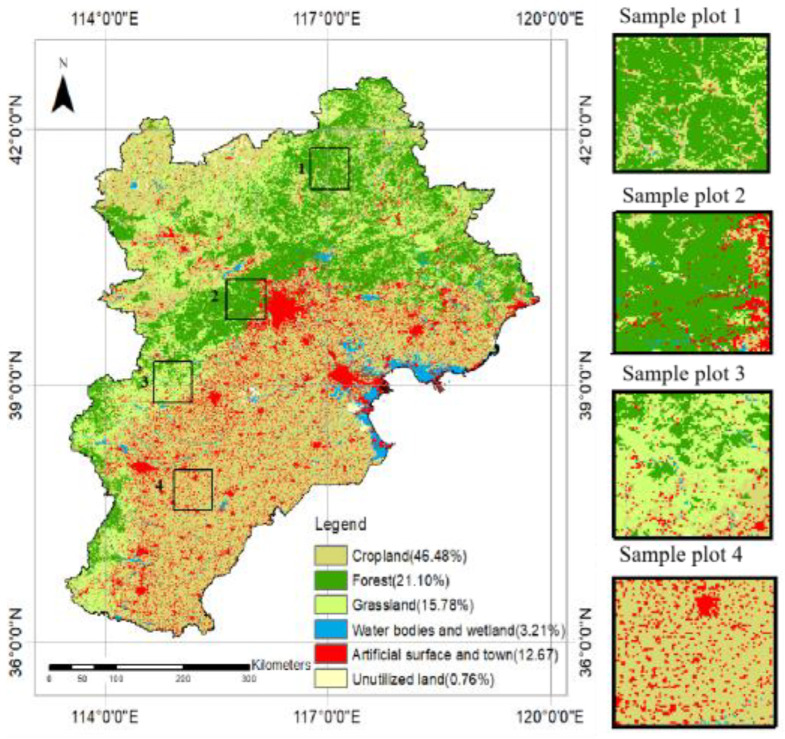
Land use types and the sample plots in the Beijing–Tianjin–Hebei region.

**Figure 2 sensors-23-02040-f002:**
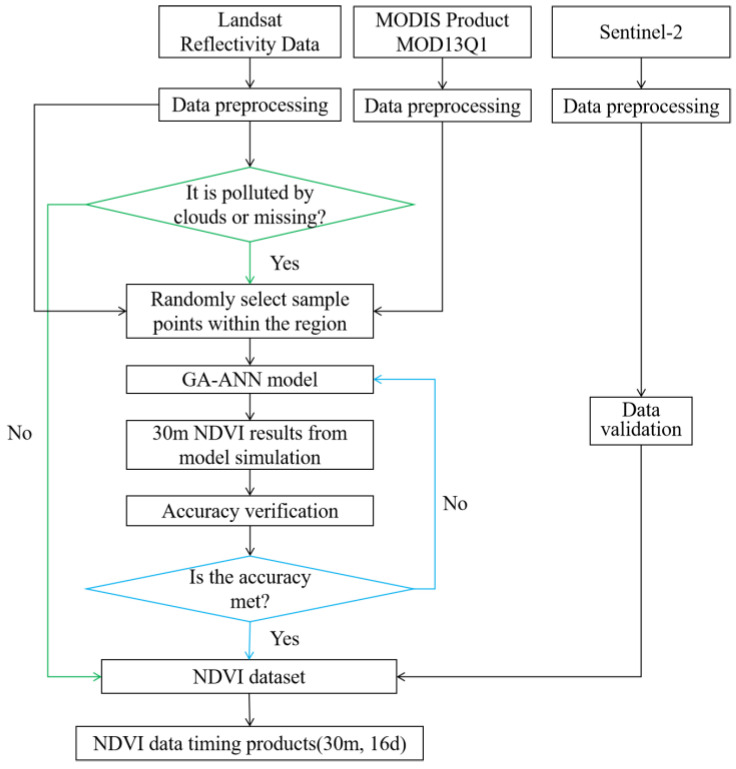
Flowchart for reconstructing the NDVI (“polluted by clouds” refers to areas covered by clouds and cloud shadows; “missing” refers to areas not covered by Landsat imagery).

**Figure 3 sensors-23-02040-f003:**
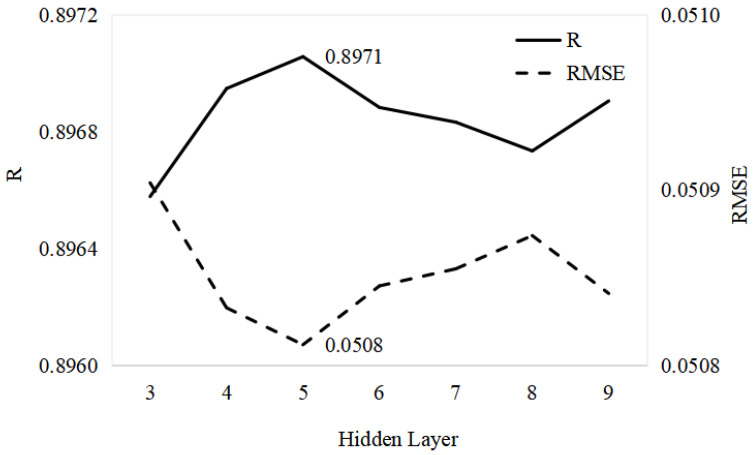
The RMSE and R of different hidden layers of the reconstructed NDVI model based on GA-ANN.

**Figure 4 sensors-23-02040-f004:**
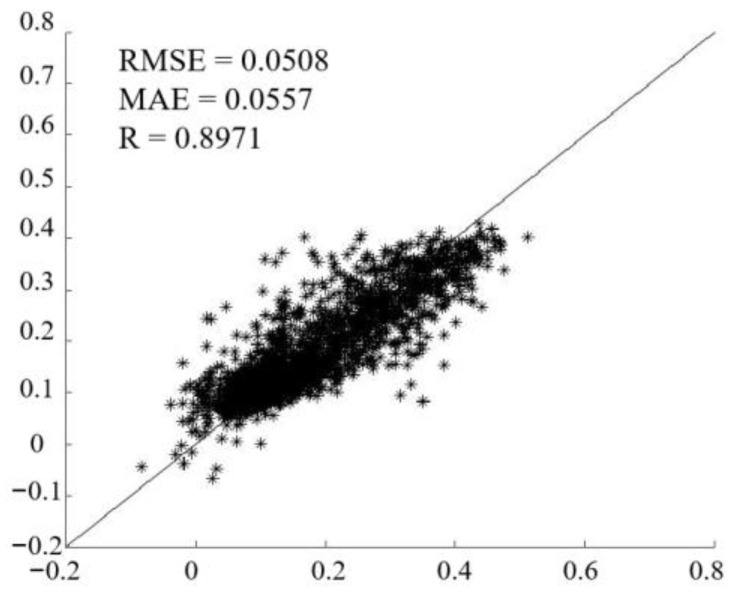
Model testing results of reconstructed NDVI model.

**Figure 5 sensors-23-02040-f005:**
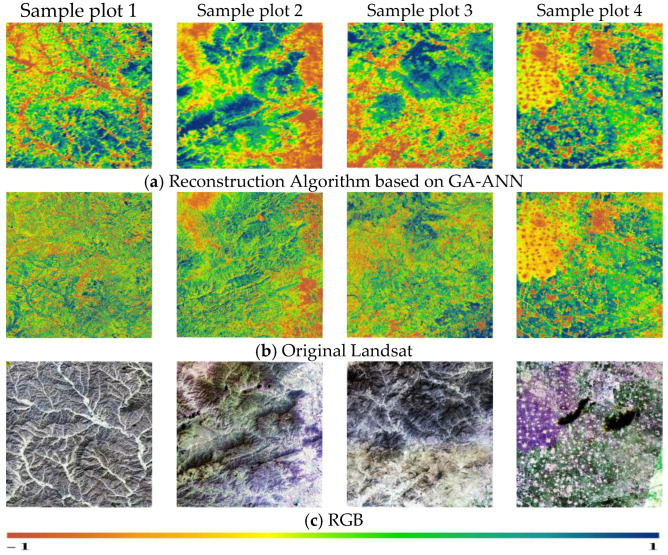
Authenticity test between reconstructed NDVI model and original Landsat NDVI.

**Figure 6 sensors-23-02040-f006:**
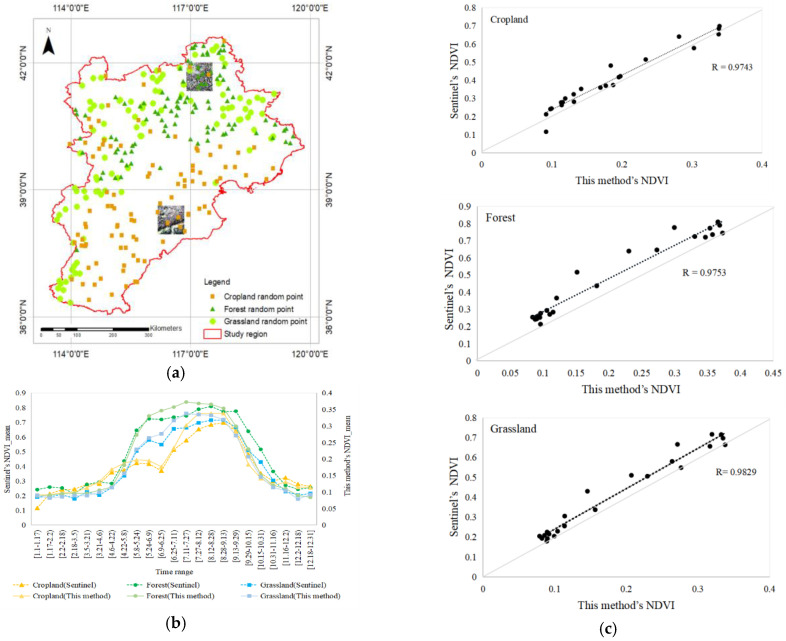
Validation of random point and true-color image distribution and comparison of the mean NDVI statistical analysis for the method in this paper and Sentinel. (**a**) Validation of random point and true-color image distribution. (**b**) Change in the mean value of NDVI. (**c**) Comparison of this paper's method with Sentinel NDVI for different vegetation types.

**Figure 7 sensors-23-02040-f007:**
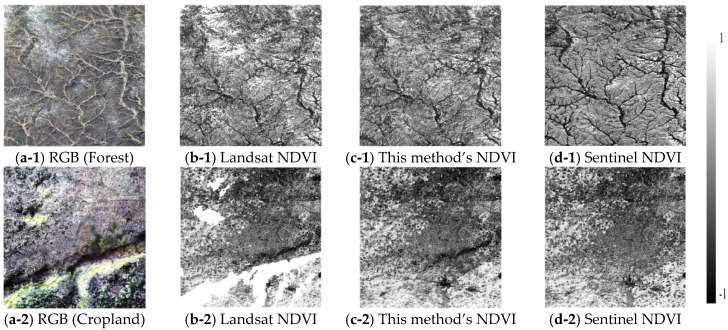
The verification of this paper’s NDVI and Sentinel NDVI from a visual point of view.

**Figure 8 sensors-23-02040-f008:**
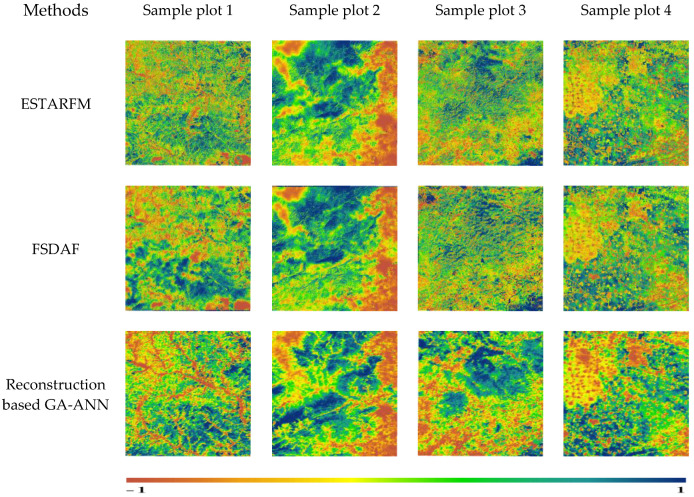
The NDVIs of ESTARFM, FSDAF, and the methods used in this study from a visual point of view.

**Figure 9 sensors-23-02040-f009:**
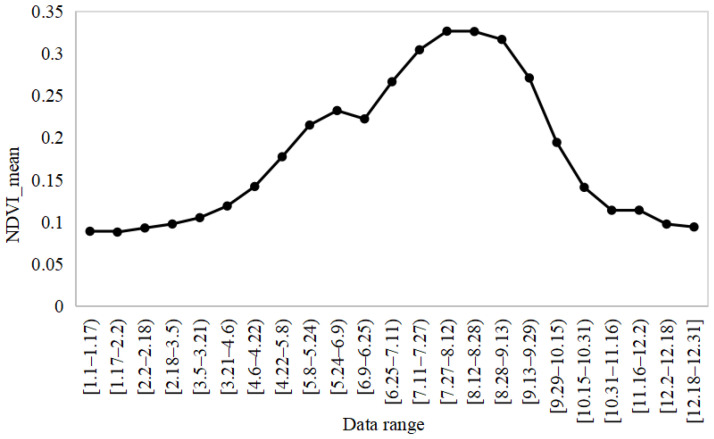
Temporal change of mean NDVI in Beijing–Tianjin–Hebei.

**Figure 10 sensors-23-02040-f010:**
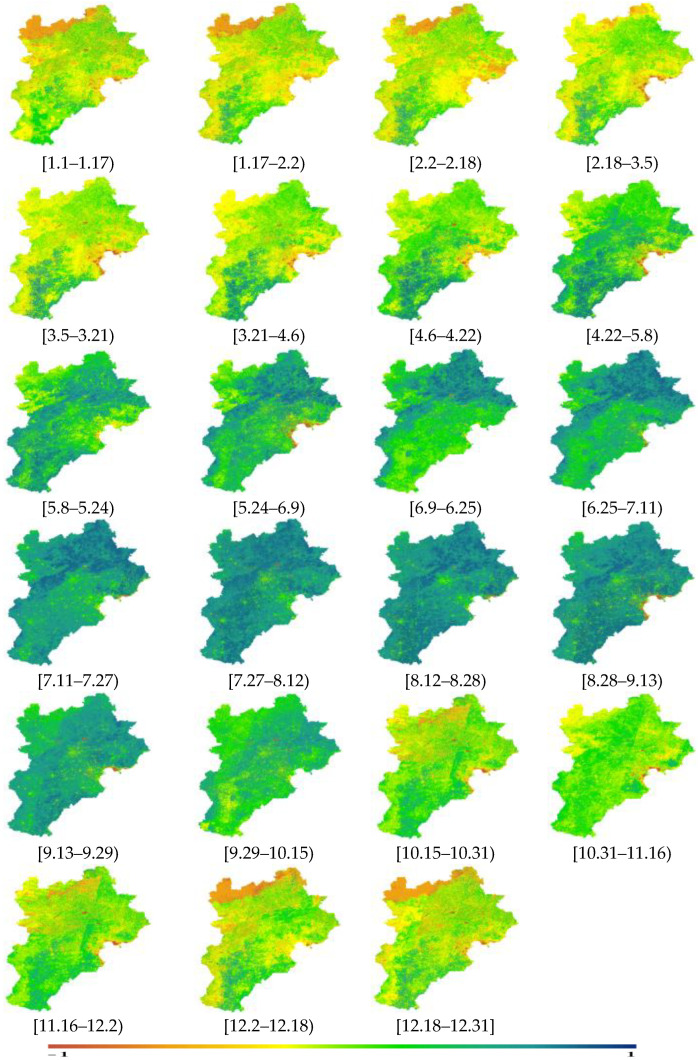
Spatial changes in NDVI for Beijing–Tianjin–Hebei.

**Figure 11 sensors-23-02040-f011:**
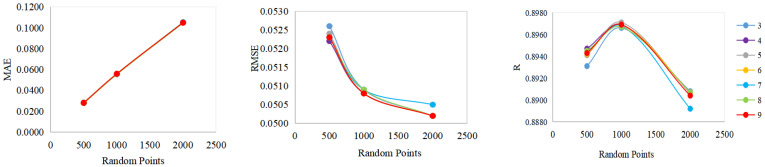
The MAE, RMSE, and R with different numbers of random points and hidden layers (3 to 9 are hidden layers).

**Figure 12 sensors-23-02040-f012:**
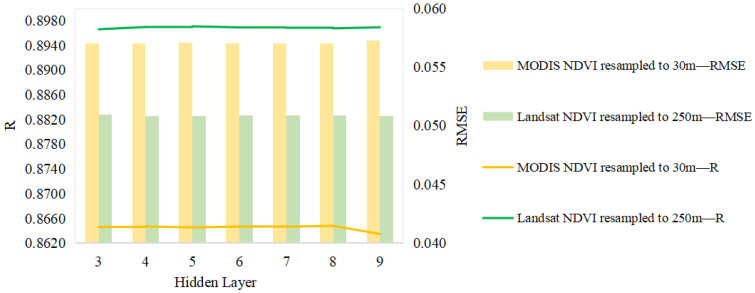
RMSE and R for two resampling methods (Landsat resampled to 250 m; MODIS resampled to 30 m).

**Table 1 sensors-23-02040-t001:** The satellite data used in this study and the spatial and temporal resolution of the reconstructed NDVI.

Data	Spatial Resolution	Temporal Resolution	Band	Time Range
MOD13Q1	250 m	16 d	NDVI	2018-01-01–2020.12.31
Landsat 8 L2	30 m	16 d	SR_B4 (RED) SR_B5 (NIR)	2018-01-01–2020.12.31
Sentinel-2 L2	10 m	5 d	B4 (RED) B8 (NIR)	
Reconstructed NDVI	30 m	16 d		

**Table 2 sensors-23-02040-t002:** NDVI calculation formulae for Landsat 8 L2 and Sentinel-2 L2.

Landsat 8 L2			Sentinel-2 L2		
Band	Bandwidth (nm)	NDVI Formula	Band	Bandwidth (nm)	NDVI Formula
4	636–673	$\mathrm{NDVI}=\frac{B5-B4}{B5+B4}$	4	664.5 nm (S2A)/665 nm (S2B)	$\mathrm{NDVI}=\frac{B8-B4}{B8+B4}$
5	851–879		8	835.1 nm (S2A)/833 nm (S2B)	

**Table 3 sensors-23-02040-t003:** Parameters of reconstructed NDVI model based on GA-ANN.

Model Parameter	Type/Value	Model Parameter	Type/Value
Population size	10	Elite count	9
Population type	Double vector	Migration direction	Forward
Population initial range	16 × 2 double	Migration interval	11
Selection mechanism	Roulette wheel	Time limit	Infinite
Basis of chromosome selection	Fitness function	Stall generation limit	Infinite
Crossover type	Double	Maximum number of generations	50
Crossover probability	0.4	Termination criteria	0.00001
Mutation type	Gaussian	Display	Iteration
Mutation probability	0.2		

**Table 4 sensors-23-02040-t004:** Accuracy assessment of reconstructed NDVI model based on GA-ANN.

Site	RMSE	MAE	PSNR	SSIM
Sample plot 1	0.0199	0.0331	15.9963	0.7838
Sample plot 2	0.0241	0.0518	15.6056	0.7024
Sample plot 3	0.0131	0.0198	20.4122	0.8679
Sample plot 4	0.0066	0.0392	16.1757	0.7632

**Table 5 sensors-23-02040-t005:** Comparison table of RMSE and MAE for each method with original Landsat NDVI.

Sites	ESTARFM		FSDAF		GA-ANN	
	RMSE	MAE	RMSE	MAE	RMSE	MAE
Sample plot 1	0.0153	0.0243	0.0437	0.0457	0.0199	0.0331
Sample plot 2	0.2317	0.2319	0.2301	0.2305	0.0241	0.0518
Sample plot 3	0.0140	0.0213	0.0120	0.0332	0.0131	0.0198
Sample plot 4	0.0346	0.0577	0.0478	0.0610	0.0066	0.0392

**Table 6 sensors-23-02040-t006:** Hidden layers and random points of reconstructed NDVI model based on GA-ANN.

Hidden Layer	Random Points								
	500			1000			2000		
	MAE	RMSE	R	MAE	RMSE	R	MAE	RMSE	R
3	0.0282	0.0526	0.8931	0.0557	0.0509	0.8966	0.1050	0.0502	0.8908
4	0.0281	0.0522	0.8947	0.0557	0.0508	0.8969	0.1051	0.0502	0.8907
5	0.0281	0.0524	0.8941	0.0557	0.0508	0.8971	0.1050	0.0502	0.8906
6	0.0281	0.0523	0.8942	0.0557	0.0508	0.8969	0.1051	0.0502	0.8907
7	0.0281	0.0523	0.8944	0.0557	0.0509	0.8968	0.1054	0.0505	0.8892
8	0.0281	0.0523	0.8945	0.0557	0.0509	0.8967	0.1050	0.0502	0.8907
9	0.0281	0.0523	0.8943	0.0560	0.0508	0.8969	0.1050	0.0502	0.8904

## Data Availability

http://ladsweb.nascom.nasa.gov/ (accessed on 11 April 2022).

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
