# Peer review of "High-Spatial-Resolution NDVI Reconstruction with GA-ANN"

_sensors, 2023, doi:10.3390/s23042040_

Round 1
Reviewer 1 Report
I have read the manuscript titled with “High Spatial Resolution NDVI Reconstruction of Long Time Series Dataset with GA-ANN” (sensors-2169369). The manuscript used the GA-ANN method to get high spatial resolution of NDVI data by using the Landsat and Modis data in 2020. Overall, the novel of methods for this manuscript is limited, and the structured should be largely improved.
The authors state that “NDVI for Landsat was not continuous due to the influence of cloud and cloud shadow”. In my opinion, it is a part of causa essentiae leading the uncontinuous time series of Landsat. Other sensors also had this drawback such as Terrra MODIS. The long revisit period is one of key factors to cause this result, just like the author's statement in Section 3.1.3. The authors said they proposed the method GA-ANN. However, GA-ANN had already been proposed by predecessors. The authors highlight their study reconstruted a long time series. However, in the manuscript, only the time-series in 2020 were recontructed.
Furthermore, I have several concerns, which I think must be addressed properly before this manuscript is suitable for publication. More detail comments can be found in the attached PFD file.

Author Response
Thanks for your suggestions. We redid the experiment and reorganized the content and structure of the article. We have made revisions to each comment, the specific content is shown in the article.

Reviewer 2 Report
Please see the attached.

Author Response
Thanks for your suggestions. We redid the experiment and reorganized the content and structure of the article. We have made revisions to each comment, the specific content is shown in the article and the attached PDF.

Reviewer 3 Report
The comments are attached in the following pdf document.

Author Response

(The authors gave the same response as above.)

Round 2
Reviewer 1 Report
After carefully review on the manuscript (sensors-2169369-v2), I think the author should further highlight their novelty of the manuscript, and Figure 2 is unnecessary. I have no more comments on this study.
Author Response
Thank you for your suggestion. Your suggestion is very helpful to us.
We have removed Figure 2 in the paper of “2.2. Dataset and Data Preprocessing” and we also reordered figure numbers.
Reviewer 2 Report
I noticed an overall improvement on text quality and it is much better now. I have no extra comments.
Author Response
Thank you for your suggestion. Your previous suggestion is very helpful to us.
We have also edited the article for long sentences and incorrect text, and marked with revision mode.
Reviewer 3 Report
Thanks for providing adequate explanations to the scientific method in the revised version along with supporting images and flowchart.
However, the manuscript requires an extensive English editing as there are numerous grammatical mistakes / sentence structural mistakes which makes the readability of the manuscript low. Please provide simple sentences with less abbreviatd terminologies to make the explanations comprehensible.
After the extensive English editing the manuscript is ready to be published.
Author Response
Thank you for your suggestion. Your suggestion is very helpful to us.
We have deleted the unnecessary abbreviations in the "1. Introduction" section.
We have also edited the article for long sentences and incorrect text, and marked with revision mode.
